# Experimental and Modeling Investigation for Slugging Pressure under Zero Net Liquid Flow in Underwater Compressed Gas Energy Storage Systems

Chengyu Liang [1], Wei Xiong [1,*], Meiling Wang [1], David S. K. Ting [2], Rupp Carriveau [2] and Zhiwen Wang [1]

1    Department of Mechanical Engineering, Dalian Maritime University, Dalian 116026, China
2    Turbulence and Energy Laboratory, University of Windsor, Windsor, ON N9B 3P4, Canada
*    Correspondence: xiongwei@dlmu.edu.cn

**Abstract:** As an emerging flexible-scale energy storage technology, underwater compressed gas energy storage (UW-CGES) is regarded as a promising energy storage option for offshore platforms, offshore renewable energy farms, islands, coastal cities, etc. Liquid accumulation often occurs in underwater gas transmission pipelines, which is a challenge to overcome. In this study, an experimental investigation is carried out on the pressure distribution characteristics of liquid accumulation flow in hilly terrain under the condition of Zero Net Liquid Flow. A slug flow pressure model with different inclination angles at four times is established and verified, and its error range is within ±20%. Analysis revealed that reduction and growth in pressure difference are related to the outflow of slug in an inclined pipe. A high-speed camera is used to capture the movement of liquid accumulation under Zero Net Liquid Flow (ZNLF) and record the associated dynamic parameters. By imaging the motion of liquid accumulation and detecting the pressure changes in the pipeline at various times, the pressure fluctuation in the pipeline at the slug flow cause is studied. Outcomes from this work can be leveraged to help further the development of underwater compressed gas energy storage technology.

**Keywords:** gas transmission; slugging pressure; zero net liquid flow; underwater compressed gas energy storage

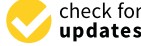



## 1. Introduction

Energy storage is a driving force for the development of renewable and sustainable energy infrastructure. Compressed air energy storage (CAES) has attracted progressively more attention in recent years [1]. Many branches of compressed air energy storage such as adiabatic CAES (A-CAES) [2–4], isothermal CAES (I-CAES) [5], isobaric CAES (I- CAES) [6], sub-critical liquid air energy storage (SC-LAES) [7], supercritical CAES (SC-CAES) [8], and underwater CAES (UW-CAES) [9,10], have been explored. UW-CAES is a flexible-scale energy storage technology that can store compressed air in deep-water air accumulators. It is particularly suitable for energy storage from offshore platforms, offshore renewable energy facilities, islands, and coastal cities [11–14]. As renewable energy grows rapidly, achieving carbon compliance and carbon neutralization is the goal of all countries [15]. This trend will further promote the rapid development of UW-CAES in the next few decades. Many researchers have started working on the structural design, system modeling, application scenarios, and energy efficiency of UW-CAES [16–20]. However, less attention has been paid to the underwater gas transmission process along the long pipelines, a key factor restricting the development of UW-CAES. In the transport of underwater gas, the transmission pipe wall temperature decreases with the increase in water depth. The heat exchange between the pipe wall and seawater causes the gas temperature in the pipe to gradually decrease, and condensation takes place when the dew point pressure is reached [21]. The liquid is formed in the pipeline and it falls due to the action of gravity, leading to liquid accumulation in the low-lying part of the pipeline [22,23]. Driven by the

flowing gas, a complex and troublesome slug flow is formed, and this adversely affects pipeline transportation, as displayed in Figure 1.

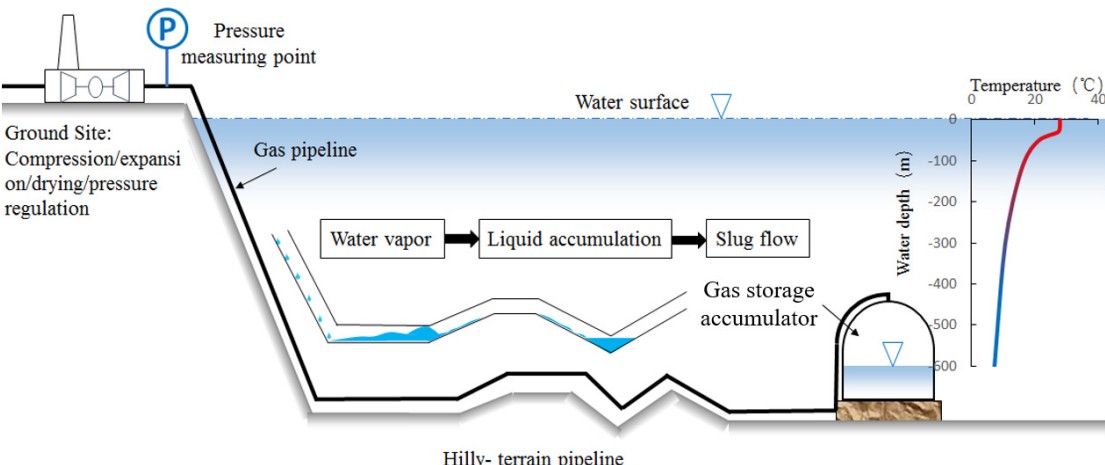

**Figure 1.** Formation of slug flow in transmission pipeline.

Liquid accumulation often forms in low–lying pipelines, and hilly terrain pipelines are the most common in pipeline transportation. Many scholars have researched the slug flow pressure in a hilly terrain pipeline [24]. This provides a reference value for the study of ZNLF in a UW-CAES hilly terrain pipeline. The conditions that are required for ZNLF to occur include the accumulation of all liquid in the lowest parts of the pipe, then a portion of the liquid is cut off by the flowing gas, and the rest circulates in the upward pipe section to form ZNLF [25,26]. Gregory et al. [27] carried out liquid accumulation flow experiments with inner diameters of 25.8 mm and 51.2 mm and inclination angles of 1°, 5°, and 9.2°. From the experimental pressure drop data, the amount of liquid accumulation in the pipe depended on the inclination angle and gas flow. Blowing out all liquid accumulation required a relatively high gas flow. In addition, the study clearly showed that the presence of liquid accumulation increases the pressure loss of the natural gas pipeline. Based on Gregory et al.'s experiment (1981), Kokal and Stanislav [28] proposed a slug flow pressure drop prediction model in the middle of an inclined pipeline under ZNLF. Experiments with inclination angles of 1°, 5°, and 9.2° were carried out on 26 mm and 51 mm pipes. Following that, Amaravadi et al. [29] performed experimental and theoretical studies on the pressure in upward pipes under ZNLF. The pressure experimental data of four upward inclination angles (1°, 2°, 5°, and 9°) were obtained. A prediction model for the average liquid holdup and pressure gradient of the liquid accumulation exit velocity under ZNLF was developed. Compared with the experimental data and model of Kokal and Stanislav (1986), Amaravadi et al.'s prediction model results were good match with the experimental data. ZNLF is a special state in the pipeline, and there are few studies on pipeline fluidity in this state. Later scholars mostly studied slug flow in pipes under general conditions. However, it also provides theoretical guidance for the pressure study of slug flow under ZNLF.

Slug flow is the most common flow pattern in pipeline transmission [30]. Numerous studies have attempted to predict the slug flow in the two-phase flow pipes. Dukler and Hubbard [31] proposed a slug flow prediction model of gas–liquid flow based on the observation. The slug body velocity, the slug head velocity, the liquid film velocity, the slug length, the liquid film area behind the slug, and the surface shape of the mixing area can be predicted. It was matched with the experimental data of Gregory and Scott [32]. Wallis [33] established the slug flow cell model of and predicted the phase fraction and pressure gradient. However, that model depends on some key assumptions and the choice of four closure rules. Then, Fabre [34] discussed these basic issues and emphasized their limitations by comparing them with the existing data. On this basis, the flow assumption of the model was explored and improved. Felizola and Shoham [35] established a unified

model of slug length and slug body liquid holdup as a function of inclination angle. In contrast to experimental data, this model can be used to predict slug flow behavior in upward pipelines.

The above slug flow models are all established based on steady-state conditions. Bendiksen et al. [36] presented a transient slug flow prediction model. The model combined the single-cell slug flow model and the numerical transient two-fluid models and applied the closure law to the description of the physical characteristics of slug flow. Following that, many studies focused on the establishment of the hydrodynamic model of slug flow. Colmenares et al. [37] placed a hydrodynamic prediction model of slug flow in a horizontal pipe. PDVSA INTEVEP experimental data were used to evaluate the model and the absolute average relative error of the model in predicting the pressure drop of slug flow was less than 6%. Zhang et al. [38] established a unified hydrodynamic model based on slug flow hydrodynamics to predict the flow pattern transition, pressure gradient, liquid holdup, and slug characteristics of gas–liquid two-phase pipe flow at different angles. The model made significant inroads in eliminating discontinuities in closed relationships. Dong et al. [39,40] established a hydrodynamic model of slug flow by a slug element. On this basis, a heat transfer model was deduced. The model was verified by experimental data of different void fraction, pressure drop, and two-phase heat transfer coefficient.

In some recent studies, the experimental and visual study of slug flow has played a prominent role. Zhu et al. [41] conducted a non-invasive measurement of slug flow vibration of a curved flexible riser. The vibration displacement and gas–liquid flow in the pipe were observed. They found that the vibration caused by the slug flow mainly occurred on the curvature surface of the flexible riser. The fluctuation of the slug flow pressure in the pipeline participated in the dynamic response of the riser, and the vibration frequency was also observed in the pressure fluctuation, which was reflective of the fluid–structure interaction. Kim and Kim [42] investigated the pressure drop characteristics of slug flow, using air and low-viscosity mineral oil in a pipe with an inner diameter of 4 cm. Combined with the visual image and pressure drop oscillation time, the relevant flow parameters related to slug flow were measured, and the theoretical slug flow model for predicting the total pressure gradient of slug unit was established by using the empirical correlation of slug flow parameters. Liu et al. [43] visualized the characteristic parameters (velocity, length, and liquid film thickness) and pressure drop of slug flow in a narrow rectangular channel. Based on the measurement of slug bubble liquid film thickness by a PCB liquid film sensor, a new correlation for predicting liquid film thickness was proposed. In addition, the existing pressure drop correlations based on separation models were summarized and evaluated using the new data. AL-Dogail et al. [44] conducted slug flow pressure experiments using air and water on a flow circuit with a diameter of one inch and a length of twenty-five feet. The pressure sensors were arranged along the pipeline direction, and the pressure response data generated by statistical analysis and regression analysis were used to characterize the slug flow. They found that the pressure drop decreased with the increase in gas flow rate.

To the best of the authors' knowledge, the pressure changes in slug flow formed by liquid accumulation movement in hilly terrain pipelines under ZNLF have not been studied. In addition, the slug pressure at different moments of the slugging pipe segment has not been analyzed. Furthermore, there was no comprehensive monitoring of liquid accumulation from movement to discharge. This paper aims at filling these gaps by invoking experiments and the establishment of a slug flow model to analyze the pressure changes at slug flow that occurs in a hilly terrain pipeline under ZNLF. First, the four-stage pressure model of slug movement in the upward inclined pipe is established, and the model validity is verified. Second, by comparing the pressure difference at different times, it is concluded that its change is related to the liquid slug outflow in the pipe. Finally, the moving visual image of the slug in the pipe from entering the inclined pipe to discharging is captured.

This paper is organized as follows: Section 2 presents the test rig and the procedure of the experiment. In Section 3, the slug flow pressure model built into this study is described. In Section 4, the results and discussion of the analyzed data are conveyed. Finally, the conclusions and expectations are provided in Section 5.

## 2. Experiment

### 2.1. Test Rig and Measurement Details

The slug flow experimental investigation was performed in a hilly terrain pipeline as presented in Figure 2. The test bench of the experimental system was composed of test section, air supply pipeline, image acquisition device, and data acquisition, as illustrated in Figure 2. More details of the facility were described as follows.

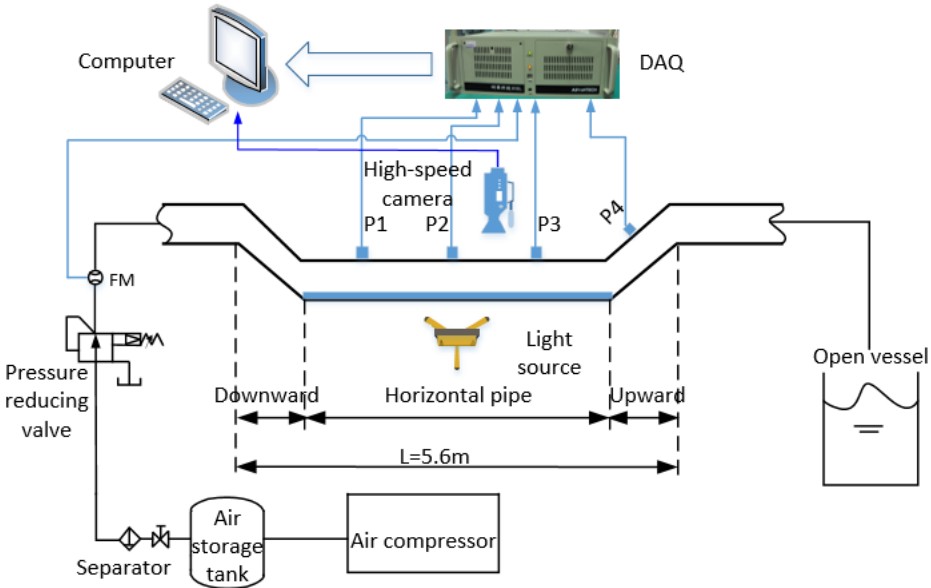

**Figure 2.** Schematic diagram of experiment.

The material of the test pipe was organic glass and the inner diameter was 26 mm, the outer diameter was 32 mm, and the total length was 6600 mm. Aluminum pipe clamps were used for supporting the horizontal pipe, the ascending inclined pipe, and the descending inclined pipe. The horizontal pipe and inclined pipe connection was implemented using the organic elbows with angles of 5°, 10°, 15°, and 20° in the test section (Figure 2). Therefore, a hilly terrain low-lying section was formed.

The compressed air flow was supplied from an Atlas Copco air compressor and an Atlas Copco tank with a capacity of 1.5 m$^3$. The gas flow rate was measured by FESTO-565406 sensor (measuring range: 0~1000 L/min) with an accuracy of ±0.3%. The filtered gas was supplied by an Atlas Copco purifier, with a high filtration efficiency. The AS4000-03 was used to adjust the inlet velocity of the pipeline from 0 to 1670 L/min. The pressure-reducing valve was subjected to the IR3020-03, with a range of 0.01 to 0.8 MPa and a sensitivity of 0.2%. Four pressure detection points were arranged on the test pipeline, three were located at the top of the horizontal pipe and one was located in the middle of the inclined pipe as presented in Figure 2. MPM489 with an accuracy of ± 0.5% (within the range of 0 to 1.6 MPa) was used to measure the pressure.

All acquisition sensors were connected to the data acquisition instrument and computer. The data acquisition system was composed of Advantech industrial computer IPC-610L and acquisition card PCI-1710U. The resolution used in the experiment was 1920 × 1020 pixel high-speed camera (50 mm f/1.8D lens) at 1500 frames per second to visualize slug flow movement. A sufficient light source guaranteed the clear recording under high-speed conditions. The temperature of the laboratory was about 25 °C, ignoring

the effects of temperature and humidity on the density and viscosity of the experimental fluid. The physical parameters of fluids were summarized in Table 1.

**Table 1.** Physical parameters of fluids.

| | Water | Air |
|---|---|---|
| Density | 997.05 kg·m$^{-3}$ | 1.184 kg·m$^{-3}$ |
| Kinetic Viscosity | $8.9008 \times 10^{-4}$ Pa·s | $1.849 \times 10^{0-5}$ Pa·s |
| Surface Tension | $\sigma_{A/W} = 0.07197$ N·m$^{-1}$ | |

*2.2. Experimental Procedure*

First, the water of various volumes, 50 mL, 80 mL, 100 mL, 200 mL, and 300 mL were injected into the pipe. Blue plant dye was used to color the liquid accumulation in the pipe to facilitate the observation of liquid slug movement. Second, the compressor was turned on and the pressure in the pipe was stabilized by adjusting the pressure-reducing valve. Then, the gas entered the pipeline, capturing the movement of the liquid in the pipe while recording pressure changes in the pipe. Next, the inlet gas velocity was adjusted, and a high-speed camera was used to track the liquid movement and the formation of slug flow. Afterward, the pipeline was cleaned and the experiments with different liquid accumulation volumes and different inclination angles were completed via the above steps. Finally, we closed the air source, and recovered and recycled the liquid used in the experiment. Through the experimental investigation of different pipe inclination angles and liquid volumes, the pressure variation in the pipe at the liquid accumulation movement in the concave pipe of ZNLF forming slug flow was obtained.

**3. Modeling**

*3.1. Slug Unit Pressure Model in Inclined Pipe*

Slug flow is extremely common in gas–liquid mixed transportation pipelines, and it is mainly dominated by liquid slug filling the pipeline section or alternating flow of large bubbles. The experimental pipeline in this study is along a hilly terrain pipe, where the liquid tends to accumulate in the lowest places to block the gas, resulting in slug flow in the up-dip pipeline.

The physical model of the inclined pipe slug unit is shown in Figure 3. A liquid slug unit ($L_u$) is divided into two areas: the liquid film area ($L_f$) and the liquid slug area ($L_s$) are displayed in Figure 3. The motion of the slug flow is related to these two areas. If the flow velocity in the liquid slug area is greater than that in the liquid film area, the liquid slug overtakes the liquid film in front, so that part of the liquid slug occupies the liquid film area. Due to gravity and the shear force between the pipe wall and the gas–liquid interphase, the liquid in the tail part of the liquid slug falls off and merges into the liquid film as it cannot catch up with the accelerated part. Under this movement, the liquid slug area is pushed forward, forming a slug flow. In a slug unit, the continuity equations of liquid and gas phases are [45]:

$$\frac{V_{LS}A\rho_L}{f} = A\rho_L\alpha_s t_s V_{ll} + A\rho_L\alpha_f t_f V_f \tag{1}$$

$$\frac{V_{GS}A\rho_G}{f} = A\rho_G(1-\alpha_s)t_s V_{lg} + A\rho_G(1-\alpha_f)t_f V_t \tag{2}$$

where $V_{LS}$ and $V_{GS}$ are the superficial velocities of liquid and gas phases in m/s. $V_{ll}$ and $V_{lg}$ are the true average velocity of liquid and gas in the liquid slug area in m/s. $\alpha_s$ and $\alpha_f$ are the liquid holdup of the liquid slug area and liquid film area. $V_m$ is the mixing velocity of liquid and gas in the liquid slug area in m/s. $V_f$ is the average velocity in the liquid film area in m/s. $V_t$ is the average bubble velocity in the liquid film area in m/s. $\rho_L$ and $\rho_G$ are the density of the liquid and the gas in kg/m$^3$, respectively. $f$ is the slug frequency in Hz.

$t_s$ and $t_f$ are the time taken for the liquid slug and bubble, respectively, to pass through a certain point in s.

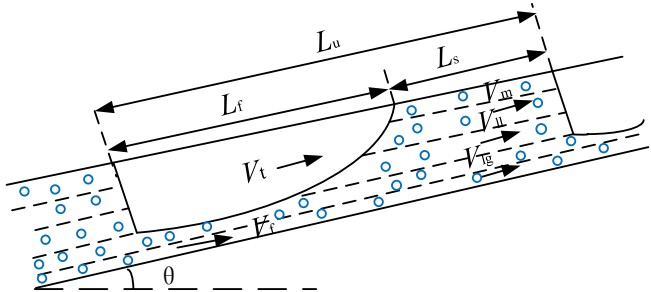

**Figure 3.** Physical model of the slug unit in inclined pipe.

Time for the liquid slug to pass through the liquid slug area and liquid film area:

$$t_s = \frac{L_s}{V_m}, \; t_f = \frac{L_f}{V_f}, \tag{3}$$

Time for bubbles to pass through the liquid slug area and liquid film area:

$$t_s = \frac{L_s}{V_t}, \; t_f = \frac{L_f}{V_t} \tag{4}$$

Substituting Equation (3) into Equation (1), and Equation (4) into Equation (2), we obtained:

$$\frac{V_{LS}}{f} = \frac{\alpha_s V_{ll} L_s}{V_m} + \alpha_f L_f \tag{5}$$

$$\frac{V_{GS}}{f} = \frac{(1 - \alpha_s) V_{lg} L_s}{V_t} + (1 - \alpha_f) L_f \tag{6}$$

where $L_s$ is the liquid slug area length in m and $L_f$ is the liquid film area length in m.

It is assumed that the slug liquid level in the liquid film area is fully developed and uniform. Under this assumption, considering the incompressibility of gas and liquid, the overall liquid phase mass balance on the slug unit is obtained:

$$V_{LS} L_u = \alpha_s V_{ll} L_s + \alpha_f V_t L_f \tag{7}$$

For a stable slug unit, the length can be known $L_u = L_s + L_f$. Combined with Equation (7), the slug flow unit length and frequency are obtained:

$$L_u V_{LS} - L_u V_t \alpha_f = \alpha_s V_{ll} L_s + \alpha_f V_t L_f - (L_s + L_f) V_t \alpha_f \tag{8}$$

$$L_u = L_s \frac{V_{ll} \alpha_s - V_t \alpha_f}{V_{LS} - V_t \alpha_f} \tag{9}$$

$L_s$ can be obtained according to Scott and Kouba [46] as:

$$\begin{cases} L_s = 30D & D < 0.038m \\ \ln L_s = -26.6 + 28.5(\ln D + 3.67)^{0.1} & D \geq 0.038m \end{cases} \tag{10}$$

$$f = 0.048 h_s V_{ll} \frac{2.02 + Fr}{D} \tag{11}$$

$$Fr = \frac{V_s^2}{gD} \tag{12}$$

where $F_r$ is the Froude number. It is expressed as a dimensionless parameter of the relative magnitude of fluid inertia force and gravity.

For the liquid phase

$$A\rho_L \alpha_s (V_t - V_{ll}) = A\rho_L \alpha_f (V_t - V_f) \tag{13}$$

After simplification, it can be arranged as:

$$\alpha_s (V_t - V_{ll}) = \alpha_f (V_t - V_f) \tag{14}$$

By analyzing Equation (14), it can be shown that on any section of a slug unit, the total volume flow is constant. For the cross-sections of the liquid slug area and the liquid film area:

$$V_m = V_{GS} + V_{LS} = V_{lg}(1 - \alpha_s) + V_{ll}\alpha_s \tag{15}$$

$$V_m = V_t(1 - \alpha_f) + V_f\alpha_f \tag{16}$$

$$V_t = CV_{ll} + V_d \tag{17}$$

In the above equation, $C$ and $V_d$ [47] can be calculated:

$$C = \begin{cases} 1.05 + 0.15 \sin^2\theta & Fr' < 3.5 \\ 1.20 & Fr' \geq 3.5 \end{cases} \tag{18}$$

$$V_d = \begin{cases} V_{dH}\cos\theta + V_{dV}\sin\theta & Fr' < 3.5 \\ V_{dV}\sin\theta & Fr' \geq 3.5 \end{cases} \tag{19}$$

$$V_{dH} = 0.54\sqrt{gD} \tag{20}$$

$$V_{dV} = 0.35\sqrt{gD} \tag{21}$$

$$Fr' = \frac{V_s}{\sqrt{gD}} \tag{22}$$

where $C$ is the proportional coefficient, $V_d$ is the bubble drift velocity in m/s; $V_{dH}$ and $V_{dV}$ are the floating velocities of bubbles in static liquid in vertical and horizontal pipelines in m/s, respectively; $\theta$ is the pipeline inclination angle in radians; D is the inner diameter of the pipe in m; and $V_m$ is the mixing speed of liquid phase and gas phase in liquid slug area in m/s.

In the above equations, the expression of the average velocity $V_f$ in the liquid film area is provided in Equation (14), the liquid true average velocity $V_{ll}$ in the liquid slug area can be calculated from Equation (15), and the bubble's average velocity $V_t$ in the liquid film area can be obtained from Equation (16). Furthermore, the slug average liquid holdup αL can be defined as:

$$\alpha_L L_u = \alpha_s L_s + \alpha_f L_f \tag{23}$$

Combining Equation (7), Equation (14), and Equation (15), it can be shown that,

$$\alpha_L = \frac{V_t\alpha_s + V_{lg}(1 - \alpha_s) - V_{GS}}{V_t} \tag{24}$$

$\alpha_s$ can be obtained as suggested by Andreussi et al. [48]:

$$\alpha_s = \frac{V_m - V_{mf}}{(V_m - V_{mo})^m} \tag{25}$$

$$m = 1 - 3(\rho_G/\rho_L) \tag{26}$$

$$V_{mf} = 2.6\sqrt{gD}\left[1 - 2\left(\frac{D_0}{D}\right)^2\right] - \frac{V_d}{C - 1} \tag{27}$$

$$V_{\text{mo}} = \frac{240}{C-1}\sqrt{E}\left(1 - \frac{1}{3}\sin\theta\right)\left[\frac{g\sigma(\rho_L - \rho_G)}{\rho_L^2}\right]^{1/4} + \frac{V_d}{C-1} \tag{28}$$

$$E = \frac{(\rho_L - \rho_G)gD^2}{\sigma} \tag{29}$$

$$D_0 = 0.025\text{m} \tag{30}$$

where $\sigma$ is the surface tension in N/m. Following Crawford et al. [49] we obtain:

$$\alpha_f = 4D_T - 4D_T^2 \tag{31}$$

$$V_{\text{lf}} = \frac{V_m - (1 - \alpha_f)V_t}{\alpha_f} \tag{32}$$

$$D_T = 0.0682 \times \left[\frac{\mu_L^2}{gD^3(\rho_L - \rho_G)\rho_L}\right]^{1/3} \times \left[\frac{4\rho_L|V_{\text{lf}}|D_T \times D}{\mu_L}\right]^{2/3} \tag{33}$$

where $V_{\text{lf}}$ is the liquid velocity in the liquid film area in m/s and $\mu_L$ is the liquid viscosity in Pa·s.

From the above equations, the liquid film area pressure drop $Dp_f$ and the liquid slug area pressure drop $Dp_s$ as shown in Figure 4 are calculated as follows:

$$Dp_s = \frac{\tau_s\pi D}{A}\frac{L_s}{L_u} \tag{34}$$

$$Dp_f = \left(\frac{\tau_f S_f + \tau_G S_G}{A}\right)\frac{L_f}{L_u} \tag{35}$$

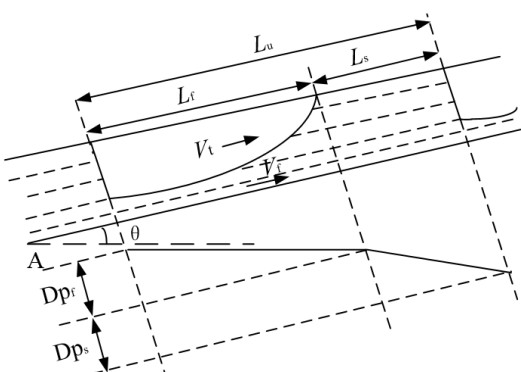

**Figure 4.** Pressure drop of slug flow unit.

The pressure drop of a slug unit is the sum of these two parts, that is:

$$Dp_u = Dp_s + Dp_f \tag{36}$$

The shear force calculation in the above formula is shown in Table 2.

### 3.2. Slug Flow Pressure Model in Inclined Pipe

The pressure drop of the slug unit can be calculated by the Section 3.1 slug unit pressure model, which is crucial for the slug flow pressure in the pipeline. The slug flow in the pipe is formed by the movement of the slug unit at different times. By calculating the pressure drop of the slug unit at different times, the movement state of the slug flow in a pipe can be divided, and the slug flow pressure model is established. The liquid slug movement in the inclined pipe is roughly divided into four stages, as presented in Figure 5.

**Table 2.** Parameter calculation method.

| Parameter | Expression |
| --- | --- |
| Gas phase wetting perimeter $S_G$ | $S_G = \pi D - S_L, (\text{m})$ |
| Liquid phase wetting perimeter $S_L$ | $S_L = \arccos(1 - 2h_L/D)D, (\text{m})$ |
| Cross sectional area of pipe A | $A = \pi D^2/4, (\text{m}^2)$ |
| Gas phase cross-sectional area $A_G$ | $A_G = \pi(1 - \alpha_L)D^2/4, (\text{m}^2)$ |
| Liquid phase cross-sectional area $A_L$ | $A_L = \pi \alpha_L D^2/4, (\text{m}^2)$ |
| Shear force of liquid film and wall $\tau_f$ | $\tau_f = f_f \rho_L V_f^2/2$ |
| Shear force of gas phase and wall $\tau_G$ | $\tau_G = f_G \rho_G V_G^2/2$ |
| Shear stress in the liquid slug area $\tau_s$ | $\tau_s = f_s \rho_s V_s^2/2$ |
| Friction coefficient of liquid film and wall $f_L$ | $f_f = C_L(D_L V_f/\mu_L)^{-n}$ |
| Friction coefficient of gas phase and wall $f_G$ | $f_G = C_G(D_G V_G/\mu_G)^{-m}$ |
| Friction factor between gas and phase in liquid slug area $f_s$ | $f_s = C_G(D V_s/\mu_s)^{-m}$ |
| Hydraulic diameter of liquid phase $D_L$ | $D_L = 4A_L/S_L$ |

For laminar flow, $C_L$, $C_G$, $m$, and $n$ are 1. For turbulent flow, $C_L$ and $C_G$ are 0.046, $m$ and $n$ are 0.2.

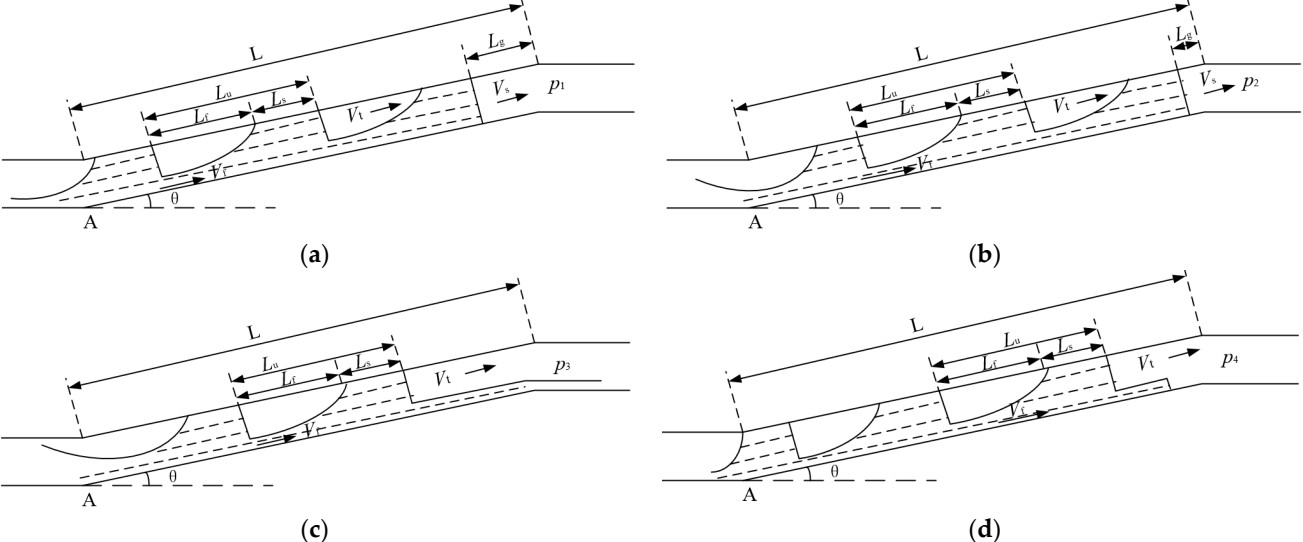

(a)　　　　　　　　　　　　　　　　　　　　　(b)

(c)　　　　　　　　　　　　　　　　　　　　　(d)

**Figure 5.** Slug flow pipeline pressure distribution: (a) $t_1$; (b) $t_2$; (c) $t_3$; (d) $t_4$.

At $t_1$, the liquid slug area of the slug unit at the elbow just enters the upward inclined pipe section, and there is no liquid backflow in the inclined pipe as presented in Figure 5a. The slug head speed $V_s$ is obtained as follows.

$$V_s = V_{LS} + V_{GS} \tag{37}$$

where $V_{LS}$ and $V_{GS}$ are the superficial velocities of liquid and gas phases, respectively.

According to the mass conservation of the liquid phase, the liquid volume in the liquid film area is $AV_{LS}/f$, which can be converted into the length of the liquid film area $AV_{LS}/(f\alpha_s)$. The length of the liquid film-free area is calculated as follows.

$$L_g = \frac{L_s V_s - AV_{LS}V_t/f\alpha_s}{V_t - V_s} \tag{38}$$

The number of slugs in the inclined pipe is obtained,

$$n_1 = \left[\frac{L - L_g}{L_u}\right] \tag{39}$$

where [] signifies rounding.

Ignoring the pressure drop caused by the gas in the pipeline in the liquid film-free area, the pressure at point A at $t_1$ is obtained.

$$p_A^{t_1} = p_1 + (n_1 + 1)Dp_u + (L - (n_1 + 1)L_u - L_g + L_f)\frac{Dp_f}{L_f} \tag{40}$$

At $t_2$, the liquid slug area at the elbow completely enters the upward inclined pipeline, and part of the liquid film area also enters this inclined passage. The first slug in the inclined pipe is about to dissipate at the top of the pipe as shown in Figure 5b. Comparing the position of the liquid slug at $t_1$ and $t_2$, the expression for $t_2$ is:

$$t_2 = t_1 + (L_g + L_s)/V_t \tag{41}$$

At this time, the number of slugs in the inclined pipe is obtained.

$$n_2 = \left[\frac{L - L_s - L_g + V_t(t_2 - t_1)}{L_u}\right] \tag{42}$$

At time $t_1$, the pressure drop caused by the gas in the pipeline in the liquid film-free area is also ignored, and the pressure at point A at $t_2$ can be expressed as follows.

$$p_A^{t_2} = p_2 + n_2 Dp_u + (L_s - (V_t - V_s)(t_2 - t_1))\frac{Dp_s}{L_s}$$
$$- (L - n_2 L_u - L_g - L_s + V_t(t_2 - t_1))\frac{Dp_f}{L_f} \tag{43}$$

At $t_3$, the first liquid slug in the inclined pipe is dissipated, and part of the liquid flows out through the top of the pipe as displayed in Figure 5c. Based on the slug position at time $t_2$, $t_3$ can be calculated.

$$t_3 = t_2 + L_u/V_t \tag{44}$$

At this time, the number of liquid slugs in the inclined pipe is obtained.

$$n_3 = \left[\frac{L}{L_u}\right] \tag{45}$$

The pressure drop caused by the gas in the pipeline in the liquid film-free area is ignored, and the pressure at point A can be expressed as:

$$p_A^{t_3} = p_3 + n_3 Dp_u + (L - n_3 L_u)\frac{Dp_f}{L_f} \tag{46}$$

At $t_4$, a new liquid slug appears at the elbow, and after the liquid slug at the top of the pipe is dissipated, partial liquid film backflow, and there is a liquid film-free area again in the inclined pipe as shown in Figure 5d. Depending on the position of the slug unit at time $t_3$, $t_4$ can be solved.

$$t_4 = t_3 + [(n_3 + 1)L_u - L]/V_t \tag{47}$$

The number of liquid slugs in the inclined pipe is obtained.

$$n_4 = \left[\frac{L + V_t(t_4 - t_3)}{L_u} - 1\right] \tag{48}$$

Ignoring the pressure drop caused by the gas in the pipeline in the liquid film-free area, the pressure at point A can be expressed as follows.

$$p_A^{t_4} = p_4 + n_4 Dp_u + \frac{Dp_s}{L_s} + (L_f - (V_t - V_f)(t_4 - t_3))\frac{Dp_f}{L_f} \tag{49}$$

## 4. Results and Discussion

The arrangement of pipeline pressure measuring points in the experiment is shown in Figure 6. The slug flow state in the pipe can be obtained by establishing the pressure model of the slug flow at the above four moments. The pressure of point A and $p_1$, $p_2$, $p_3$, and $p_4$ in the slug flow pressure model established in Section 3 can be measured in the experiment. Therefore, the pressure difference between point A and $p_1$, $p_2$, $p_3$, and $p_4$ at different times is used as the method of model verification. According to the pressure difference between measuring point A at different times and the measuring point on the inclined pipe, the state of slug flow in the pipe can be judged, which has a guiding significance for identifying the flow in the pipe. To verify the correctness of the slug flow pressure model, the model value is calculated and compared with the measured pressure value.

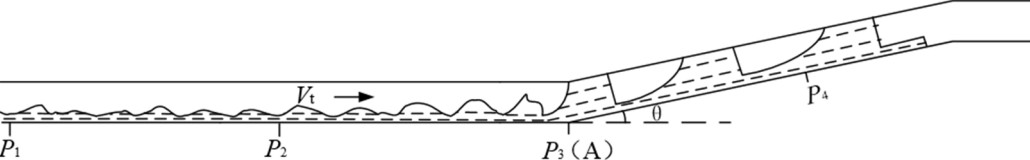

**Figure 6.** Pipeline pressure detection distribution.

In the experiment, at the gas velocity is 9.89 m/s and an inclination of 5°, slug flow does not occur when the liquid accumulation is 50 mL. However, at the same speed and angle, the slug flow occurs when the liquid accumulation volume is 80 mL, 100 mL, 200 mL, and 300 mL. The experimental and model calculation results of slug flow pressure difference under different liquid accumulation conditions are displayed in Figure 7.

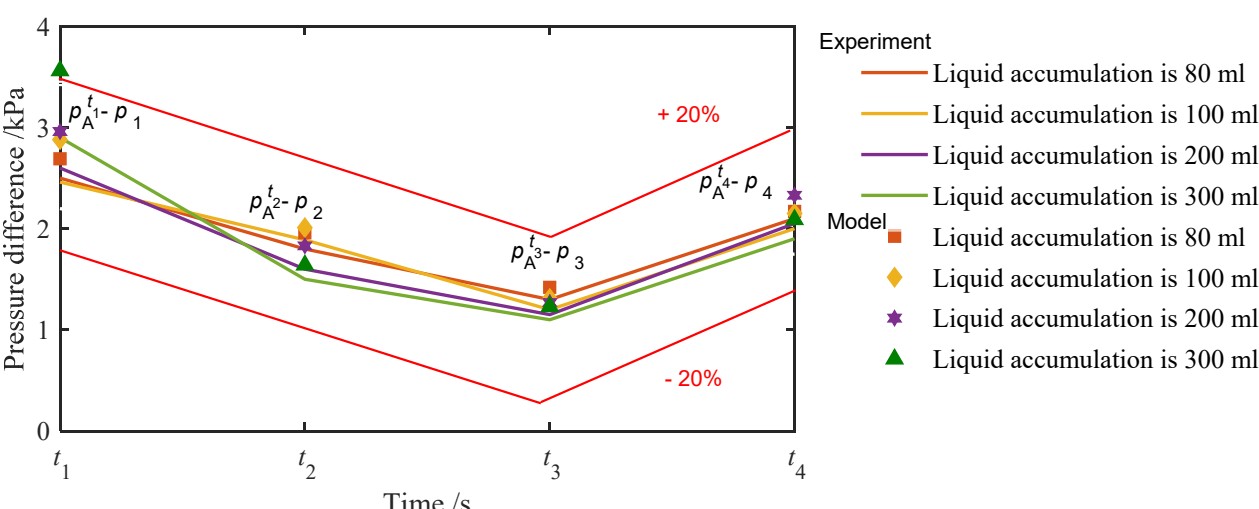

**Figure 7.** Verification of slug flow pressure model with an inclination of 5°.

It can be seen from Figure 7 that the error between the experimental and model calculated values is within ±20%. In general, the calculated value of the model is slightly larger than the experimental value. The main reason is that the slug flow completes a periodic movement continuously and rapidly in the experiment, and it is difficult to accurately divide every moment, which leads to the deviation of the measured pressure value. The secondary cause is the error caused by sensor measurement and experimental equipment.

The pressure difference in Figure 7 is divided into two stages, falling and rising. The pressure difference from t1 to $t_3$ is decreased, the reason is that the elbow is blocked by liquid accumulation, and the pressure at point A increases sharply due to the gas being compressed. While the pressure of the inclined pipe does not change much because it is connected to the outside, so the pressure difference is large.

Figure 8a–d are the slug flow movement images with the liquid accumulation of 300 mL. It can be seen that the liquid accumulation volume at point A decreases gradually from $t_1$ to $t_3$, while the liquid accumulation volume at point $P_4$ increases from $t_1$ to $t_2$, and decreases to the minimum at $t_3$. When the moment is $t_3$, the slug in the pipe is partially discharged, and the gas in the pipe is interlinked at this time, so the pressure difference at $t_3$ is the smallest. With the increase in the pressure difference from $t_3$ to $t_4$, the backflow of the liquid in the pipe makes the liquid accumulation pressure at point A increase again. However, the direction of the liquid backflow in the inclined pipe is opposite to the gas, and the impact of the two fluids causes the pressure to rise, which makes the pressure difference of $t_4$ less than $t_1$. At $t_4$, the liquid accumulation at A accumulates again, and the fluid amount at $P_4$ is also more than $t_3$, as shown in Figure 8d.

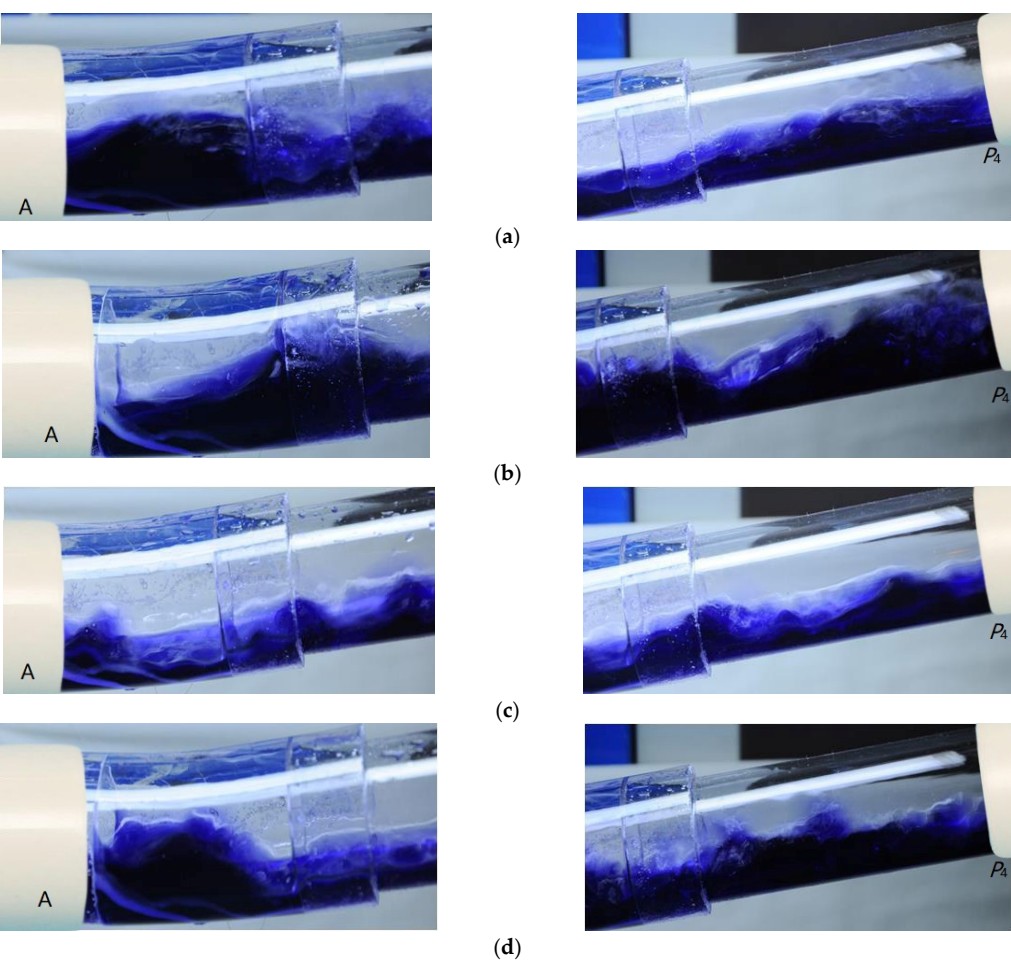

**Figure 8.** Slug flow pipeline pressure distribution 5°: (**a**) flow image in the pipeline at point A and $P_4$ at time $t_1$; (**b**) flow image in the pipeline at point A and $P_4$ at time $t_2$; (**c**) flow image in the pipeline at point A and $P_4$ at time $t_3$; (**d**) flow image in the pipeline at point A and $P_4$ at time $t_4$.

As the inclination of the pipe increases to 10°, the error range of the pressure difference between the experiment and the model calculation is also within ±20%. The trend of pressure difference is also divided into two stages, falling first and then rising as shown in Figure 9. The pressure difference of 10° has little change compared with that of 5°. As with the angle of 5°, at the gas velocity of 9.89 m/s, the liquid accumulation of 50 mL in the pipe does not cause slug flow. Slug flow in $t_1$ to $t_3$, $t_3$ to $t_4$ completed a periodic movement, from slug entering the inclined pipe to slug discharging, and then to a new slug entering.

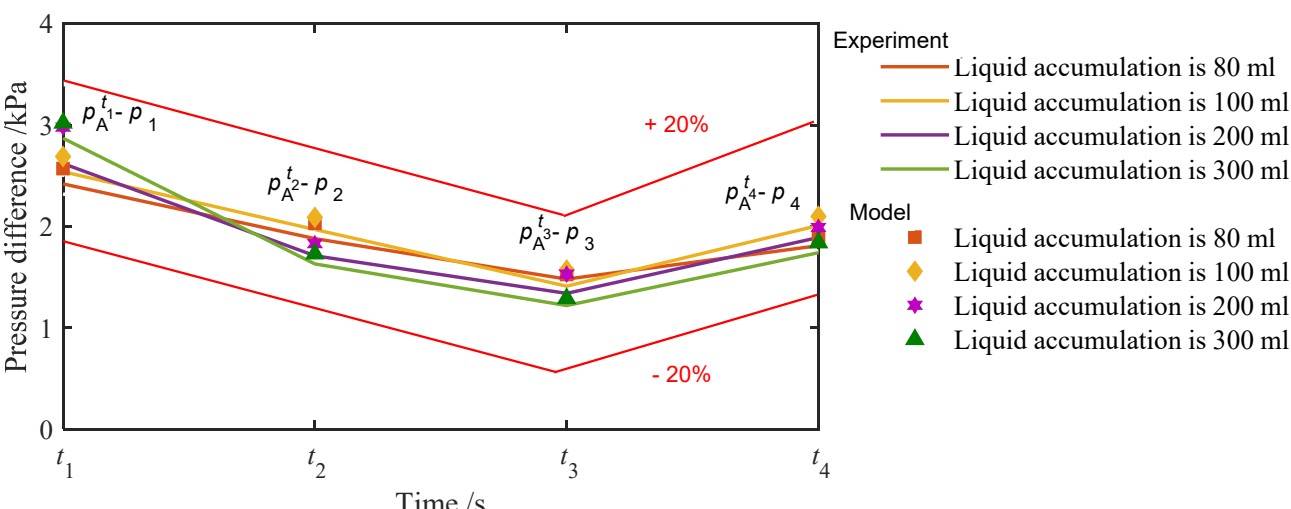

**Figure 9.** Verification of slug flow pressure model with an inclination of 10°.

When the gas velocity is 9.89 m/s, the slug flow image with the liquid accumulation of 300 mL is shown in Figure 10. At $t_1$, the liquid at point A gathers to form a liquid bridge, making preparations for the slug head to enter the pipeline, as displayed in Figure 10a. At $t_2$, the head of the slug entered the upward inclined pipe, as with the model of $t_2$ established in Section 3. The liquid film area of the slug is at point A, while the upper liquid slug in the inclined pipe begins to dissipate, as presented in Figure 10b. At $t_3$, part of the liquid flows out of the inclined pipe, and the remaining liquid flows back along the pipe, as shown in Figure 10c. The liquid re-aggregates at point A in $t_4$, as exhibited in Figure 10d. In addition, due to the direction of the liquid backflow being opposite to the gas in the inclined pipe, a small liquid slug appears in the convective extrusion.

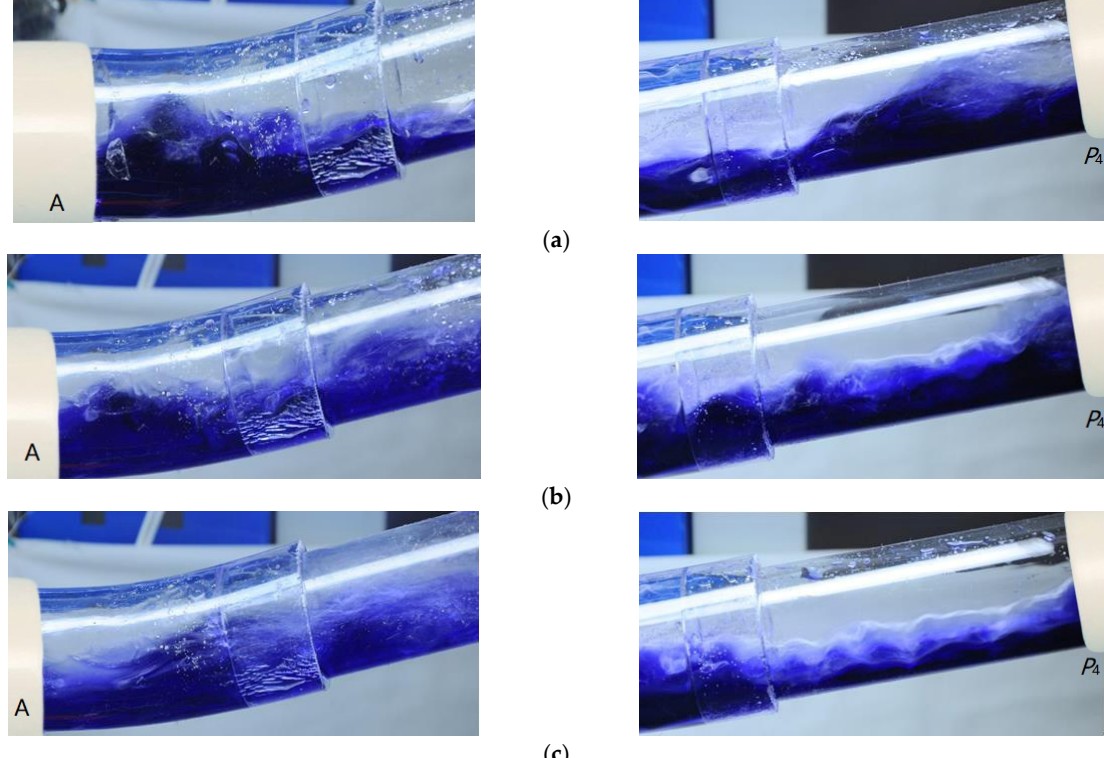

**Figure 10.** *Cont*.

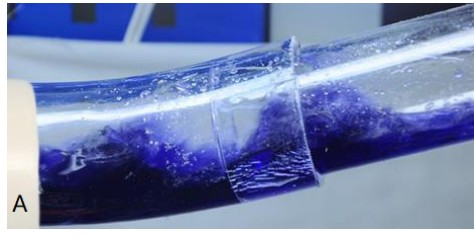
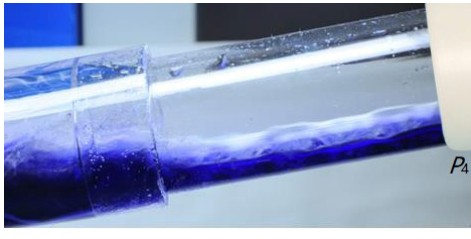

(**d**)

**Figure 10.** Slug flow pipeline pressure distribution 10°: (**a**) flow image in the pipeline at point A and $P_4$ at time $t_1$; (**b**) flow image in the pipeline at point A and $P_4$ at time $t_2$; (**c**) flow image in the pipeline at point A and $P_4$ at time $t_3$; (**d**) flow image in the pipeline at point A and $P_4$ at time $t_4$.

When the angle is 15° and the gas velocity is 9.89 m/s, the liquid accumulation volume in the pipe are 50 mL and 80 mL without slugging flow, and the deviation between the experiment and the model is within ±20%, as displayed in Figure 11. Compared with 5° and 10°, the pressure difference line slope of different liquid accumulation volume in 15° pipe is larger, indicating that the pressure change at point A and $P_4$ are more violent due to the increase in the angle. As with the above two angles, the pressure difference of slug flow in the pipe decreases first and then increases, which corresponds to the slug flow cycle of enter, discharge, and re-enter.

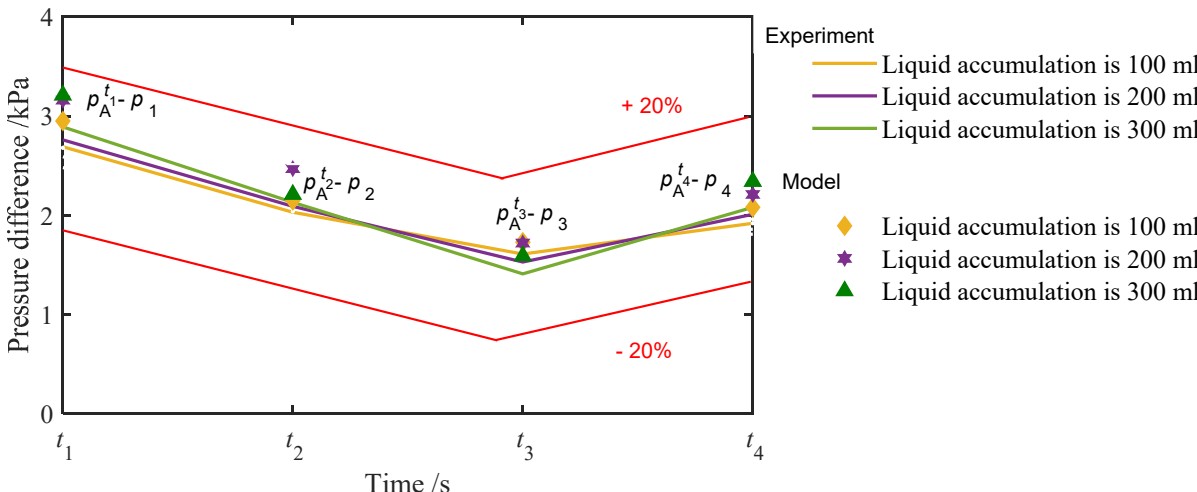

**Figure 11.** Verification of slug flow pressure model with an inclination of 15°.

Figure 12a–d are the slug flow images in a liquid accumulation of 300 mL pipe with an inclination angle of 15° and a gas velocity of 9.89 m/s. From $t_1$ to $t_4$, the liquid accumulates twice at point A and flows out once in the inclined pipe. The movement form is consistent with the pressure model established in Section 3.

The variation in the pressure difference at an inclination of 20° in Figure 13 can be divided into descending and ascending processes. Moreover, the occurrence of these two processes is associated with the outflow of slug in the inclined pipe. It can be seen from the experiment that when the liquid volume is 50 mL, 80 mL, and 100 mL, slug flow does not occur in the pipe with an inclination of 20°. Comparing the experiment and model calculation results at this angle, the error range is within ±20%, as shown in Figure 12. During the descent, part of the liquid slug is discharged from the pipeline, which reduces the blockage in the inclined pipeline and makes the pressure difference smaller. In the ascending process, due to the inclination of the pipeline, some liquid slugs flow back under gravity and collide with the gas. This increases the pressure at $P_4$ and then the pressure difference is less than that at $t_1$.

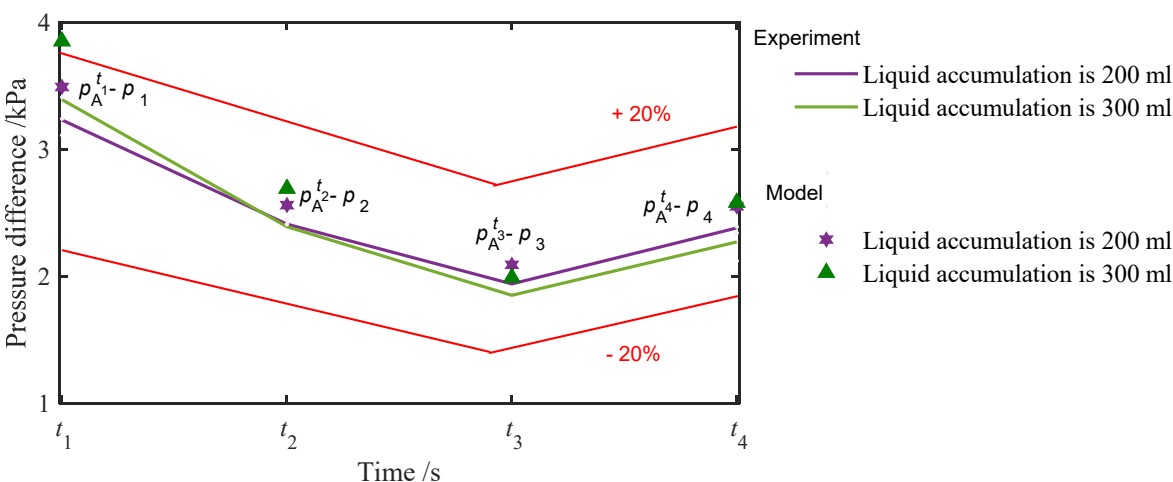

**Figure 12.** Slug flow pipeline pressure distribution 15°: (**a**) flow image in the pipeline at point A and $P_4$ at time $t_1$; (**b**) flow image in the pipeline at point A and $P_4$ at time $t_2$; (**c**) flow image in the pipeline at point A and $P_4$ at time $t_3$; (**d**) flow image in the pipeline at point A and $P_4$ at time $t_4$.

**Figure 13.** Verification of slug flow pressure model with an inclination of 20°.

When the liquid accumulation volume is 300 mL and the gas velocity is 9.89 m/s, the slug flow in the pipeline with an inclination angle of 20° forms a cycle in four stages from $t_1$ to $t_4$. That is, upon completion of $t_4$, $t_1$ reoccurs, followed by $t_2$ and then $t_3$, as displayed in Figure 14a–d. With the backflow of the dissipative liquid slug and the formation of a new liquid slug at the elbow position, the liquid accumulation in the pipe returns to the state at time $t_1$. However, the slug outflow process is the result of multiple cycles. The slug flow is rapidly discharged in this cycle until the liquid accumulation is insufficient to overcome the gravity and the interaction between the two phases, and eventually accumulates at the elbow of the pipe.

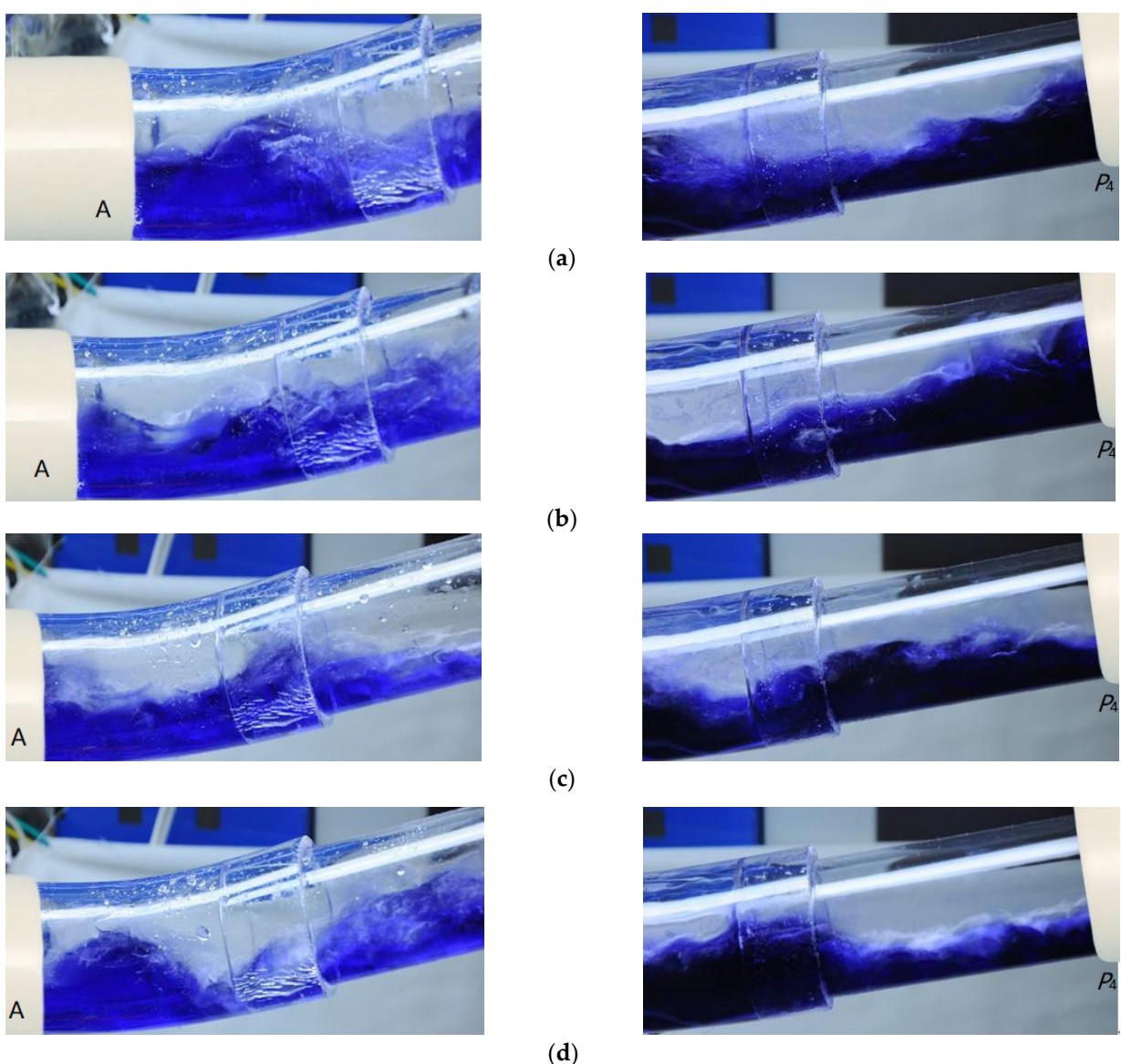

**Figure 14.** Slug flow pipeline pressure distribution 20°: (**a**) flow image in the pipeline at point A and $P_4$ at time $t_1$; (**b**) flow image in the pipeline at point A and $P_4$ at time $t_2$; (**c**) flow image in the pipeline at point A and $P_4$ at time $t_3$; (**d**) flow image in the pipeline at point A and $P_4$ at time $t_4$.

## 5. Conclusions

To effectively judge the flow state of slug flow in the pipeline, a pressure model of slug flow at different times is established and verified by calculation, which is of great significance for identifying the flow in the pipeline. An experimental investigation for slugging pressure under ZNLF was conducted with air and water as working fluids in a

26 mm diameter pipe. In the established model, the slug flow movement state is analyzed according to the pressure difference between measuring point A and $P_4$. By analyzing the pressure difference between the experiment and the model, it can be obtained that the pressure difference decreases first and then increases, which is related to the discharge of slug flow in the inclined pipe. The movement of liquid accumulation under the action of ZNLF is photographed by a high-speed camera. By studying the model and experiment, the movement of liquid accumulation in the pipe can be accurately described, and the pressure of slug flow in the inclined pipeline can be quantitatively calculated. These results provide valuable guidance for the further research and development of underwater compressed gas energy storage system and other similar systems.

**Author Contributions:** Conceptualization, C.L., Z.W. and W.X.; methodology, C.L., M.W., D.S.K.T., R.C. and W.X.; formal analysis, C.L., M.W. and Z.W.; investigation, C.L., M.W. and Z.W.; resources, W.X.; writing-original draft preparation, C.L. and M.W.; writing-review and editing, D.S.K.T., R.C., Z.W. and W.X.; visualization, C.L. and M.W.; data-curation, C.L. and M.W.; supervision, W.X.; funding acquisition, W.X. and Z.W. All authors have read and agreed to the published version of the manuscript.

**Funding:** This research was funded by National Natural Science Foundation of China, grant number 51905066 and 52075065.

**Institutional Review Board Statement:** Not applicable.

**Informed Consent Statement:** Not applicable.

**Data Availability Statement:** Not applicable.

**Conflicts of Interest:** The authors declare no conflict of interest.

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
