# Peer review of "Experimental and Modeling Investigation for Slugging Pressure under Zero Net Liquid Flow in Underwater Compressed Gas Energy Storage Systems"

_applsci, doi:10.3390/app13021216_

Round 1

Reviewer 1 Report

This paper describes models of the fluid accumulation in gas-transport pipes and its effect on the transport of the gas. Having read this paper, I believe it is of interest to the readers and will be suitable for publication after two issues have been dealt with.

The first is purely technical: The internal references to equations don't work and give the reader: Error! Reference source not found. into Eq.
Error! Reference source not found.
  Please fix this.

The second is more of an academic nature: To what extent do the results depend on the molecular structure of the gas and fluid? Can we extrapolate these results to other substances?

The focus of the reseach is the effect of fluid build-up in pipelines that are used for underwater gas storage. The authors investigate the extent to which such fluid interferes with the flow of gas through  the pipes. Research such as this is complementary, rather than innovative. However, it is highly relevant and provides much-needed information, both to other scientists, as well as the engineers that have to put this kind of knowledge into practice. As far as I could tell, the work in this particular paper is original. It has the added benefit that the paper is written in a very clear style, taking  the reader step by step through the process. Anyone who reads this paper should be capable of duplicating the research in order to either check the results, or add to them. The paper answers the question that it investigates. As I indicated in my report, I would like for trh authors to discuss the result in a broader context. As it is, the authors look at one particular setup, and I personally would like to know the extent to which we can extraoplate these results to related situations. Though, one could argue that this would be material for one or more separate papers.

Reviewer 2 Report

This work is interesting and, most important stuff, numerical results are scientific sound supported by experimental measurement. Therefore, I think the topic is suitable for the journal. However, there are several things which must be clarified more:

1) a lot of equations are not correctly referenced: Error! Reference source not found. This makes the reading and understanding more difficult. 

2) More references must be provided. For example reference for Eq.1 & 2.

3) In Eq.12), authors use Fr-number. It would be better to give one sentence talking about its physical meaning.

4) For Eq.40, it is stated that "Ignoring the pressure drop caused by the gas in the pipeline". Could authors explain whether this ignorance is important or this would bring some model errors?

5) how large is the experimental uncertainty?

Round 2

Reviewer 2 Report

Thanks a lot to authors giving effort to clarify my concerns. I agree with all response.